# Identification of a Circulating miRNA Signature to Stratify Acute Respiratory Distress Syndrome Patients

**DOI:** 10.3390/jpm11010015

**Published:** 2020-12-27

**Authors:** Gennaro Martucci, Antonio Arcadipane, Fabio Tuzzolino, Giovanna Occhipinti, Giovanna Panarello, Claudia Carcione, Eleonora Bonicolini, Chiara Vitiello, Roberto Lorusso, Pier Giulio Conaldi, Vitale Miceli

**Affiliations:** 1Anesthesia and Intensive Care Department, IRCCS-ISMETT, 90133 Palermo, Italy; gmartucci@ismett.edu (G.M.); gocchipinti@ismett.edu (G.O.); gpanarello@ismett.edu (G.P.); ebonicolini@ismett.edu (E.B.); cvitiello@ismett.edu (C.V.); 2Research Department, IRCCS-ISMETT, 90133 Palermo, Italy; ftuzzolino@ismett.edu (F.T.); pgconaldi@ismett.edu (P.G.C.); vmiceli@ismett.edu (V.M.); 3Fondazione Ri.MED, 90133 Palermo, Italy; ccarcione@fondazionerimed.com; 4Cardio-Thoracic Surgery Department Heart and Vascular Centre, Maastricht University Medical Centre, 6229 HX Maastricht, The Netherlands; roberto.lorussobs@gmail.com; 5Cardiovascular Research Institute Maastricht (CARIM), 6229HX Maastricht, The Netherlands

**Keywords:** ARDS subphenotypes, miRNAs, miRNA signature, lung injury, inflammation

## Abstract

There is a need to improve acute respiratory distress syndrome (ARDS) diagnosis and management, particularly with extracorporeal membrane oxygenation (ECMO), and different biomarkers have been tested to implement a precision-focused approach. We included ARDS patients on veno-venous (V-V) ECMO in a prospective observational pilot study. Blood samples were obtained before cannulation, and screened for the expression of 754 circulating microRNA (miRNAs) using high-throughput qPCR and hierarchical cluster analysis. The miRNet database was used to predict target genes of deregulated miRNAs, and the DIANA tool was used to identify significant enrichment pathways. A hierarchical cluster of 229 miRNAs (identified after quality control screening) produced a clear separation of 11 patients into two groups: considering the baseline SAPS II, SOFA, and RESP score cluster A (*n* = 6) showed higher severity compared to cluster B (*n* = 5); *p* values < 0.05. After analysis of differentially expressed miRNAs between the two clusters, 95 deregulated miRNAs were identified, and reduced to 13 by in silico analysis. These miRNAs target genes implicated in tissue remodeling, immune system, and blood coagulation pathways. The blood levels of 13 miRNAs are altered in severe ARDS. Further investigations will have to match miRNA results with inflammatory biomarkers and clinical data.

## 1. Introduction

Acute respiratory distress syndrome (ARDS) is an acute inflammatory lung disease characterized by the loss of lung endothelial barrier integrity, and invasion of alveoli by fluids, proteins, and inflammatory cells [1]. It is an evolutive picture that is schematically divided into an exudative phase followed by or overlapping with a proliferative phase [2]. Thus, ARDS is characterized by persistent injury stimuli and the failure of lung tissue repair that evolves into chronic lung repair, resulting in marked changes in lung structure and function [3]. In this clinical picture, the Berlin definition [4] has helped to define the syndrome, and the subsequent LUNGSAFE study described its application worldwide [5]. Nevertheless, ARDS still comprises a number of different etiologies, and likely, also different biological patterns integrating differently in the same syndrome, with potentially different responses to therapeutic actions and outcomes. The appearance of COVID-19 has cast into question the presence of a “classical” ARDS and a “new” form, with the same clinical definition but different pathophysiology [6,7,8].

To overcome this controversy, in recent years, several biomarkers have been explored to differentiate these patients [9,10,11]. These markers are not defined tools for clinical use, but interestingly, Calfee at al. have identified two ARDS subphenotypes (hyperinflammatory and non-hyperinflammatory), with different management needs and outcomes [12]. However, in such cohorts, there is a high variability of severity, which is usually measured by specific severity scores designed for critically ill patients, such as the sequential organ failure assessment (SOFA) score and the Simplified Acute Physiology Score II (SAPS II), or effectively designed for extracorporeal membrane oxygenation (ECMO) patients, such as the Respiratory ECMO Survival Prediction Score (RESP score), and the Predicting Death For Severe ARDS on VV ECMO (PRESERVE) score.

In the most severe cases of ARDS, ECMO, more frequently in its veno-venous (V-V) configuration, is the only rescue therapy able to gain time and hasten the evolution of the lung healing processes [13,14]. However, ECMO presents a new, confounding problem: contact of the blood with an artificial surface (despite advancements in biocompatible materials) can prompt the activation of coagulation factors, with further stimulation of inflammatory and tissue repair pathways [15,16].

Considering that in ECMO (a resource-consuming support from both an economic and human workload perspective), mortality is still high, new biomarkers, posed in the direction of precision medicine, may help in characterizing the heterogeneous ARDS population to discriminate groups with different prognoses, and that could be adopted to follow the disease progression, and look at the effects of the treatment given.

From a biopathological point of view, ARDS is characterized by cellular injury pathways, such as endothelial and epithelial injury, pro-inflammatory injury, coagulation, fibrosis, and apoptosis [17]. Pathophysiological differences in these processes in different ARDS severity statuses can likely be used to better understand the role of various biomarkers in ARDS. In this light, microRNAs (miRNAs) are a class of small non-coding RNA (19–25 nucleotides) responsible for silencing specific genes able to modulate specific pathways. Circulating miRNAs likely play important roles in cellular communication regulating gene expression and the phenotype of the recipient cells [18]. Moreover, in the presence of harmful stimuli, the composition of blood miRNAs can be altered, making them excellent candidates as biomarkers [19,20]. Indeed, miRNAs have been proposed as sensor-biomarkers in physiological processes such as muscular hypertrophy [21]. Otherwise, several miRNAs have been described as biomarkers for cardiovascular disease [22] and cancer [23], thus indicating that they might also play a role as signals of ongoing pathological processes.

Therefore, we aimed to explore whether the identification of a circulating miRNA signature can lead to discovery of new biomolecules useful for further characterizing ARDS patients in terms of severity status. Thus, we screened the expression of 754 circulating miRNAs in the blood of ARDS patients using high-throughput quantitative PCR to identify differentially expressed miRNAs.

## 2. Materials and Methods

### 2.1. Patient Recruitment, Characteristics, and Sample Acquisition

This study was approved by the IRCCS-ISMETT Ethics Committee (EC Code: IRRB/34/19) and was conducted in accordance with the ethical standards laid down in the 1964 Declaration of Helsinki and its later amendments. Written informed consent was initially obtained from the next of kin and later confirmed directly by the patients if recovered. Patients were enrolled if they were affected with ARDS and supported by V-V ECMO. In July 2018, we implemented miRNA measurement for ARDS patients, and subsequently the patients were enrolled in the present proof-of-concept study. Patients were categorized for anthropometric data (age, weight, and height) and severity of disease (SAPS II, SOFA score, creatinine, and bilirubin). Moreover, they were also categorized for PaO2/FiO2 ratio, specific ECMO related scores like the PRESERVE score and RESP score, which has inverse values with respect to the survival probability—lower or negative values indicate higher severity and lower probability of survival—all the scores are proportional to the severity—higher values correspond to higher severity), and pre-ECMO length of stay (LOS) in the hospital and in the intensive care unit (ICU), as well as the duration of mechanical ventilation (MV). During the ECMO run the need for continuous renal replacement therapy (CRRT) and the length of ECMO course were assessed, together with ICU- and hospital-related survival. Venous whole blood was collected in PAXgene tubes (Qiagen, Hilden, Germany), and unprocessed samples were immediately stored at −80 °C until further analysis. In addition, blood samples were collected to evaluate the following markers: creatinine, bilirubin, hematocrit, and platelets.

### 2.2. Real-Time PCR Analysis of miRNAs by TaqMan Low Density Arrays

MiRNAs were profiled using the TaqMan Array Human MicroRNA panels A and B v3.0 according to the manufacturer’s instructions (Thermo Fisher Scientific, Waltham, MA, USA). Total RNA was extracted with RiboPure™ RNA Purification Kit, blood (Thermo Fisher Scientific, Waltham, MA, USA). The purity and quantity of isolated RNA were determined by OD260/280 using a NanoDrop ND-1000 Spectrophotometer (Thermo Fisher Scientific, Waltham, MA, USA). Then, 300 ng of RNA were reverse-transcribed with the high capacity RNA-to-cDNA kit protocol (Thermo Fisher Scientific, Waltham, MA, USA) in order to produce single-stranded cDNA. QRT-PCR of 754 human miRNAs was done with the Applied Biosystems 7900 HT Real-Time PCR system, and for each miRNA, the initial data output was in SDS software v2.4. The expression level was determined with the equation 2^−ΔΔCT^ using the U6 as housekeeping gene. Furthermore, hierarchical cluster analysis of miRNA expression was used to group patients with a similar expression pattern. MiRNA expression data were grouped using Euclidean distance algorithms in the Cluster 3.0 program, and a heat map was generated using the Java TreeView program.

### 2.3. Target Gene Prediction

The miRNet [24] online database (https://www.mirnet.ca/faces/home.xhtml) was used to examine the pathways in which differentially expressed miRNAs were implicated. The database is a comprehensive atlas of miRNA-target interactions that can integrate the information resulting from 11 existing miRNA-target prediction programs (TarBase, miRTarBase, miRecords, miRanda, miR2Disease, HMDD, PhenomiR, SM2miR, PharmacomiR, EpimiR, and starBase). The software uses standard enrichment analysis based on the hypergeometric tests after adjustment for false discovery rate (FDR). In this study, to investigate the functional implications of miRNA deregulation in ARDS, we generated with miRNet pathway annotations a protein-protein interaction network of molecules targeted by at least three of our deregulated miRNAs, which are directed toward at least three genes.

### 2.4. Gene Ontology (GO) and Kyoto Encyclopedia of Genes and Genomes (KEGG) Analysis

Enrichment of the deregulated miRNAs in the biological processes GO terms was performed using the online annotation tool mirPath v.3 on the DIANA Web site (http://snf-515788.vm.okeanos.grnet.gr/). In addition, the same miRNAs were analyzed using the KEGG database on the DIANA Web site. The statistical significance threshold level for both GO enrichment and KEGG pathway analyses was *p* < 0.05.

### 2.5. Statistical Analysis

Baseline demographic and clinical characteristics of patients, as well as the blood markers during ECMO are reported as medians and 25th and 75th percentiles for continuous data and frequencies, and percentages for categorical data. All miRNA data are expressed as mean ± SD. Data from different clusters were compared using computerized statistical software with the ANOVA test. The data were further analyzed with Dunnett’s t-test. Differences were considered statistically significant at *p* < 0.05. To detect differences in miRNA expression with respect to the severity of ARDS, miRNA expression was compared between two groups (cluster A and B, which differed significantly for SAPS II, SOFA, and RESP score) using the two-tailed Mann–Whitney U test. Both the sensitivity and specificity of miRNAs to discriminate between the two clusters were assessed with receiver operating characteristic (ROC) curve analysis. All confidence intervals are reported at 95%. All tests were two-sided, and a *p* value of <0.05 was considered indicative of statistical significance. Data handling and analyses were done with SAS 9.4 software (SAS Institute Inc., Cary, NC, USA).

## 3. Results

### 3.1. Patient Characteristics, and Stratification of ARDS Severity by miRNA Expression

From July 2018 to March 2019, 13 patients were supported by V-V ECMO for ARDS at our institute. In two cases the blood samples deteriorated during the laboratory phase, thereby leaving 11 patients available for adequate blood samples miRNAs analyses. Characteristics of the overall population are presented in Table 1. Causes of ARDS were bacterial pneumonia (*n* = 6), viral pneumonia due to H1N1 Influenza A (*n* = 4), and severe polytrauma with pulmonary contusion and super-infection with bacteria in one case.

To identify deregulated miRNAs associated with ARDS severity status, starting with a panel of 754 analyzed miRNAs, we identified 229 miRNAs after quality control screening (Figure 1). Among them, hierarchical clustering analysis showed systematic variations in the miRNA expression between two different groups, and produced a clear separation of patients into two clusters (Figure 2a). Cluster A, *n* = 6, comprises patients with higher critical illness severity at baseline, while cluster B, *n* = 5, with lower baseline severity, considering the baseline SAPS 2, SOFA, and RESP score, as well as the length of ICU LOS and duration of mechanical ventilation prior to the ECMO run (Table 1, *p* < 0.05). Furthermore, as expected by the disease severity, outcomes related to LOS, such as ECMO duration or hospital LOS, show a trend over a longer stay in cluster A, without reaching statistical significance (Table 1).

Then, after the screening, using volcano plot analysis (Figure 2b) for the deregulated genes comparing the two clusters, 95 deregulated miRNAs were identified, and notably, among them, 94 were upregulated, and one was downregulated in cluster A compared to cluster B (Figure 2c,d).

### 3.2. MiRNA Target Network Construction

Starting from the 95 deregulated miRNAs, through an in silico analysis, we analyzed the association between ARDS-related pathways (regulation of tissue remodeling, regulation of immune system, and regulation of blood coagulation) and deregulated miRNAs from the miRNet database, using miRNA-target interactions and functional associations through network-based analysis. 

The 95 deregulated miRNAs were reduced to 13, and among them, the same 11 miRNAs (miR-93, -99a, -92a, -212, -20a, -19b, -18a, -17, -126, -106b, and -769) target genes implicated in all the pathways analyzed, miR-19a target genes implicated in both tissue remodeling and immune system pathways, whereas let-7a was exclusively implicated in the blood coagulation pathways (Figure 3 and Table 2). 

Furthermore, many genes had a potential relationship with both each other and with other miRNAs, as shown in Figure 3 and Table 3.

For better visualization, we defined the proteins that are likely to interact with at least *n* = 3 miRNAs. These thirteen miRNAs (miR-93, -92a, -769, -99a, -212, -20a, -19b, -18a, -17, -126, -106b, -19a, and let-7a) were considered as candidate biomarkers, and subjected to further analysis. QRT-PCR showed that these miRNAs were significantly up-regulated from 5- (miR-126, *p* = 0.04) to 163- (miR-212, *p* = 0.0006) fold in cluster A compared to cluster B (Figure 4a). Interestingly, hierarchical clustering analysis of these miRNAs revealed the same grouping (clusters A and B) (Figure 4b) already observed in the first cluster analysis of the initial 229 miRNAs (Figure 2a).

### 3.3. Enrichment of the Deregulated miRNAs in Biological Processes

We performed both GO enrichment and KEGG pathways analysis with the DIANA database. In the biological process category, among the top 30 GO terms (Figure 4c), all thirteen miRNAs were related to crucial processes potentially involved in the pathogenesis of ARDS. In particular, different terms (e.g., “cellular nitrogen compound metabolic process,” biosynthetic process,” “response to stress,” “EGFR signaling pathway,” and “cell death” terms) are involved in the regulation of tissue remodeling. Furthermore, “blood coagulation” and “platelet activation” terms were involved in the regulation of coagulation, while “immune system process” and “innate immune response” terms were involved in the regulation of the immune system. The most enriched KEGG pathways comprising terms also involved in the aforementioned biological processes are summarized in Figure 4d.

### 3.4. Deregulated miRNAs Are Associated with the Severity of ARDS Patients

To explore the clinical relevance of the deregulated miRNAs in ARDS, the thirteen miRNAs were subjected to clinical characteristics based on publicly available online applications. Consistent with the qRT-PCR results, significantly higher expression levels of thirteen miRNAs were observed. In particular, miR-93, -99a, -92a,-212, -20a, -19b, -18a, -17, -126, -19a, and let-7a were significantly up-regulated (*p* < 0.05) in ARDS patients with high severity (cluster A) compared to patients with low severity (cluster B). Instead, for miR-106b and miR-769, though the up-regulation trend was consistent with the qRT-PCR data, the difference in expression level did not reach statistical significance (Figure 5).

In order to test whether clusters A and B were categorized by common indicators to evaluate ARDS patients’ clinical illness condition, we analyzed SAPS II, SOFA, and RESP score by ROC analysis, showing that the AUC value was 0.85 (*p* = 0.02), 0.83 (*p* = 0.01) and 0.86 (*p* = 0.006), respectively (Table 4). 

Additionally, the ability of the thirteen miRNAs (miR-93, -99a,-92a, -212, -20a, -19b, -18a, -17, -126, -106b, -769,-19a and let-7a) to categorize patients with low and high severity was studied. The ROC analysis revealed that the AUC values for individual miRNAs were predictive for distinguishing between low and high severity, with an AUC ranging between 0.80 and 0.96 (Table 5).

## 4. Discussion

In this pilot study, we screened 754 circulating miRNAs extracted from the blood of ARDS patients before ECMO cannulation. We found a select pattern of miRNA expression capable of separating patients into two clusters that differ in baseline ARDS severity, and between these clusters, we performed an analysis of the differentially expressed miRNAs.

The current literature sheds light on different biomarkers to characterize ARDS patients based on the hypothesis that there are distinct sub-classes within a broader group of patients included in the same clinical definition. In fact, ARDS, as a syndrome, does not have a clearly effective treatment, if we do not consider specific treatments with low volume ventilation, the use of neuromuscular blockade, and prone position that have been demonstrated to reduce mortality [25,26,27]. Furthermore, the findings of different randomized controlled trials or non-randomized interventional trials that administer the hypothesized effective treatment to a cohort of clinically defined patients, did not reach statistical significance [28,29,30]. One possible explanation is the different response to treatment found in patients with different activated biological pathways. In this framework, to increase the knowledge of basic biological patterns in ARDS patients will be fundamental to test future treatments, even the more advanced ones.

The main achievement in ARDS studies in recent years was the definition of the “inflammatory” and “non-inflammatory” subphenotypes [31]. Recently, this concept has also been tested efficaciously on COVID-19 patients, but the approach, also with combined markers (biological and clinical, like vasopressors), still lacks complete biological explanation [31]. One element that can be explored relatively easily is miRNAs, small non-coding RNA able to modulate specific pathways. Differently from other studies, we attempted to associate miRNA expression with the severity of disease, potentially contributing to defining the prognosis, as well as evaluate treatments put in place. Our approach started from the miRNA clusterization, and from this point on defined the two cluster of patients differently for the baseline severity characteristics. This approach, was able to define the cluster of miRNA without biases due to the status of the patient, and in our opinion, has strengthened the results.

Currently, knowledge of miRNAs in ARDS is still very preliminary, but miRNAs have been demonstrated to play a relevant role in both physiological and pathophysiological human processes. Different studies have shown that the levels of certain circulating miRNAs involved in inflammation, angiogenesis, and cardiac muscle contractility are modified by the intensity and length of exercising, thus indicating that they might play a role in specific physiological processes [32,33]. Furthermore, Zhou et al. described miRNAs as potential biomarkers for cardiovascular disease [22], while Lawrie et al. first utilized miRNAs as biomarkers for cancer, in 2008 [23]. MiRNAs also play an important role in the pathogenesis of lung diseases, including ARDS [34,35,36,37,38].

To analyze the potential relationship between miRNAs and ARDS, we focused on three main processes potentially involved in ARDS pathogenesis: regulation of tissue remodeling, regulation of the immune system, and regulation of blood coagulation [9]. Accordingly, we analyzed the association between these pathways and miRNAs, showing that 13 deregulated miRNAs (miR-93, -99a, -92a, -212, -20a, -19a, -19b, -18a, -17, -126, -106b, -769, and let-7a) target genes implicated in the aforementioned pathways (Figure 3 and Table 2). Interestingly, these 13 miRNAs were previously implicated in ARDS [34,35,36,39,40], and various reports have shown that these miRNAs clearly infuence cell cycle progression/proliferation (which can potentially regulate tissue remodeling) [41,42,43,44], inflammation [45,46], and coagulation processes [47,48]. These data were confirmed by GO enrichment and KEGG pathway analysis (Figure 4c,d). Figure 5 shows that these miRNAs were up-regulated in cluster A compared to cluster B, though there were no statistical differences for both miR-106b-5p and miR-769-5p when analyzed with Mann–Whitney U test (*p* > 0.05). Furthermore, using ROC analysis, our investigation of specificity and sensitivity revealed that these miRNAs (though miR-106b-5p is not significant, *p* > 0.05) are predictive for distinguishing between low and high ARDS severity, with the AUC values for each individual miRNA ranging between 0.80 and 0.96 (Table 5). Furthermore, hierarchical clustering analysis of these miRNAs revealed the same grouping (cluster A and B) observed after the first screening, confirming the same data with a different analysis. These results just confirmed that high expression levels of miR-93, -99a, -92a, -212, -20a, 19a,-19b, -18a, -17, -126, -106b,-769, and let-7a were potentially associated with higher severity of ARDS. Our results indicate that miRNAs are promising biomolecules potentially useful for improving diagnosis, and better stratifying ARDS patients. This hypothesis is supported by the fact that different and independent studies provide evidence for the involvement of miRNAs in dysregulated ARDS signaling pathways [35,38,49,50,51], being implicated in endothelial and/or epithelial cell function, as well as in the regulation of inflammatory responses [52,53,54,55,56,57,58,59,60], and in the regulation of coagulation [61,62,63,64,65].

Our focus on the baseline values allowed us to categorize patients independently of the outcomes, which is important in recognizing the effective existence of different subgroups. Interestingly, the strength of our results resides in the fact that, with all the caveats of a limited number of cases, our patients were effectively divided into clusters.

However, we also acknowledge the limitations of our study: first, the study population was small despite being part of a peculiar setting of severity (ECMO patients) and female gender is poorly represented; consequently, the miRNAs may be biased by the high prevalence of males; second, in the period when miRNA assessment was available, ECMO was particularly effective; consequently, in terms of survival, this group of patients might not be completely representative of the general V-V ECMO population. Moreover, our results were only based on 229 (out of 754) miRNAs that passed stringent QC criteria. It is possible that some miRNAs that did not pass QC are functionally related to ARDS. Finally, ARDS is considered to be a complicated syndrome with multiple etiologies, so a single or a few miRNAs might not evidence strong signals for all ARDS patients.

## 5. Conclusions

This pilot study shows that the blood levels of 13 miRNAs, strongly related to biological pathways possibly associated with ARDS, are altered in patients with severe ARDS, and may offer diagnostic value as well as contribute to ARDS stratification. Further investigations will have to match miRNA results with inflammatory biomarkers and clinical data to confirm these preliminary observations.

## Figures and Tables

**Figure 1 jpm-11-00015-f001:**
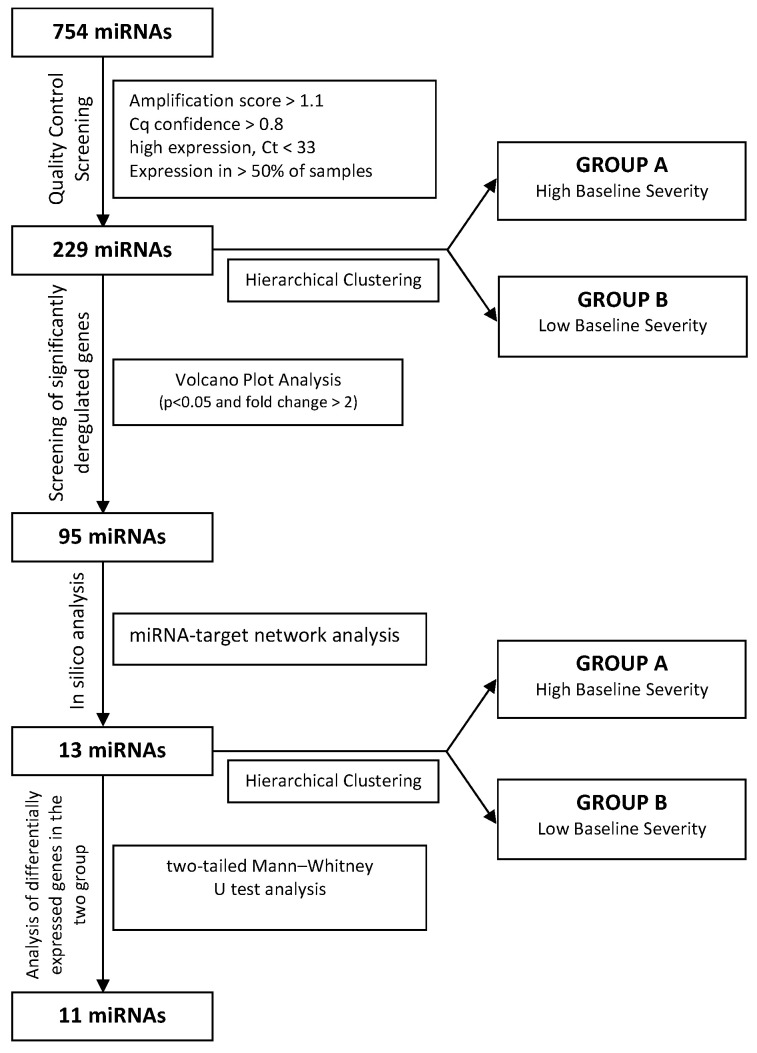
Flow chart illustrating the steps for quality control screening of miRNA expression.

**Figure 2 jpm-11-00015-f002:**
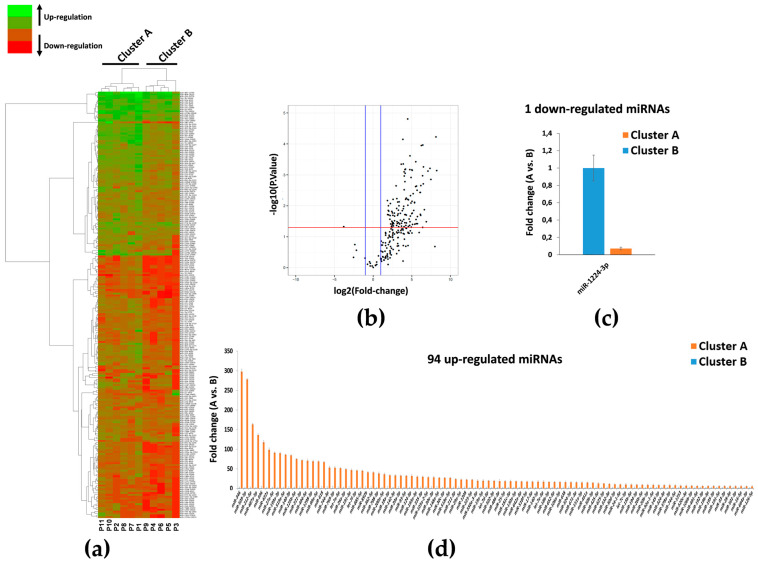
QRT-PCR analysis of miRNAs in ARDS patients. (**a**) Hierarchical clustering based on miRNA expression levels of 229 miRNAs in ARDS patients. Heatmap colors represent relative miRNA expression normalized to housekeeping. (**b**) Volcano plot analysis of deregulated miRNAs analyzed between patients in cluster A and patients in cluster B (*p* ≤ 0.05 and fold change ≥2). (**c**) Down-regulated miRNAs in cluster A patients vs. cluster B patients. (**d**) Up-regulated miRNAs in cluster A patients vs. cluster B patients. MiRNA levels were normalized to those of U6. Data are means ± SD.

**Figure 3 jpm-11-00015-f003:**
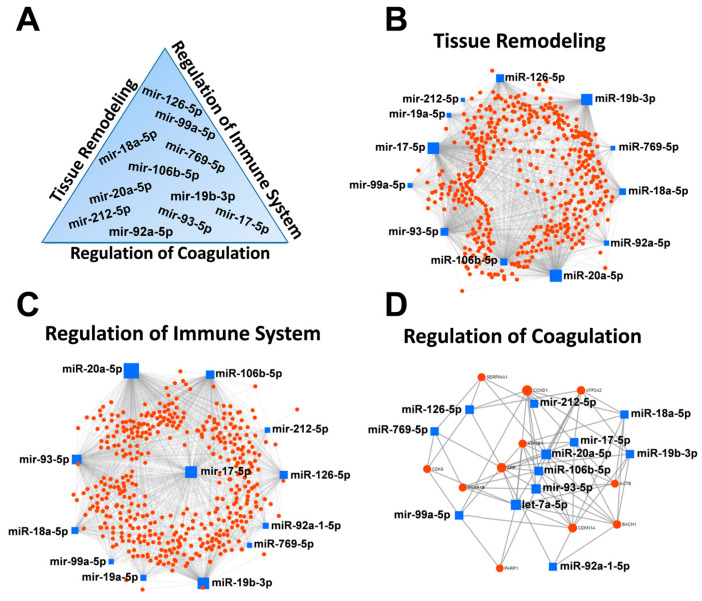
Protein-protein interaction network generated for shared miRNA target genes after miRNET analysis. Only genes targeted by at least three differentially expressed miRNAs are shown. (**A**) The top ranked deregulated miRNAs belonging to the three analyzed pathways. Images show networks with all interactions between deregulated miRNAs and genes involved in (**B**) tissue remodeling, (**C**) regulation of immune system, and (**D**) regulation of coagulation.

**Figure 4 jpm-11-00015-f004:**
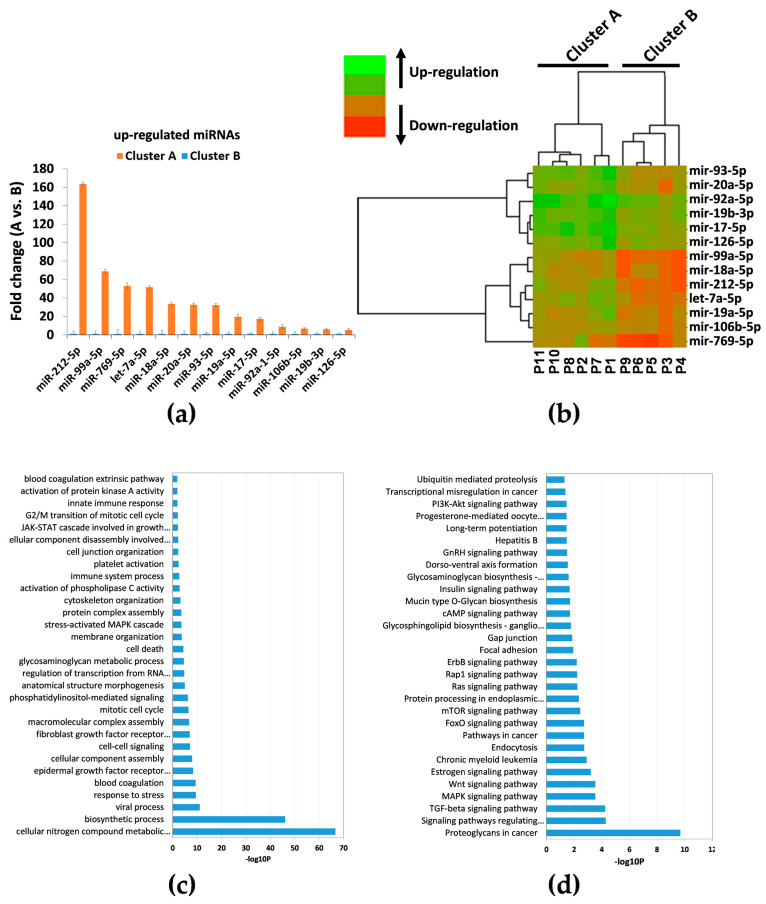
QRT-PCR, GO and KEGG analysis of thirteen miRNAs up-regulated in cluster A vs. cluster B. (**a**) Up-regulated miRNAs. (**b**) Hierarchical clustering based on miRNA expression levels of 13 miRNAs in ARDS patients. Heatmap colors represent relative miRNA expression normalized to housekeeping. (**c**) GO (partial list of biological process) enrichment analysis of thirteen up-regulated miRNA. (**d**) KEGG pathways (partial list) enrichment analysis of thirteen up-regulated miRNA.

**Figure 5 jpm-11-00015-f005:**
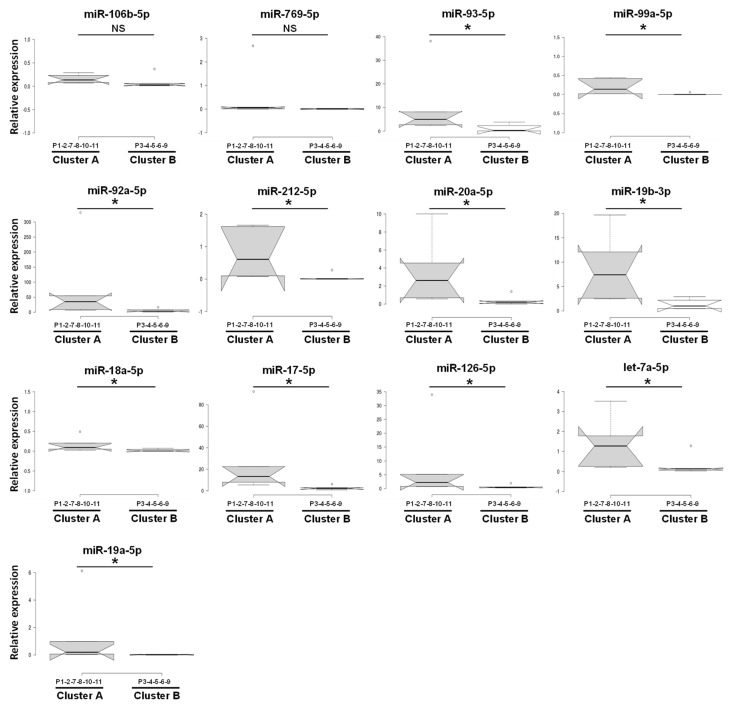
Blood expression levels of thirteen miRNAs in cluster A and cluster B. Data presented as expression relative to U6 (internal control). Box plots are displayed where the horizontal bar represents the median, the box represents the IQR, and the whiskers represent the maximum and minimum values. Comparisons made by Mann–Whitney U test. * *p* < 0.05.

**Table 1 jpm-11-00015-t001:** Patient demographics and clinical characteristics.

Variable	Overall	Cluster A (*n* = 6)	Cluster B (*n* = 5)	*p* Value (Cluster A vs. B)
Gender, N (%)	Male 9 (82%)	Male, 5 (83%)	Male, 4 (80%)	-
Age (years)	54 (44, 59)	51 (44, 59.5)	54 (53.5, 57)	0.84
Weight (Kg)	84 (68.5, 86.5)	80.5 (64.25, 84.75)	86 (83, 99.75)	0.48
Height (cm)	170 (163, 178)	168 (161.5, 173)	178 (175, 178)	0.53
BMI (Kg/m^2^)	26.6 (24.6, 28.4)	26.4 (23.8, 27.42)	28.4 (27.47, 32.4)	0.58
SAPS II	36 (33, 51.5)	51.5 (40.5, 59.5)	35 (33.5, 36)	0.03
SOFA score	5 (3.5, 9)	9 (5.75, 10)	3.5 (3, 4.5)	0.03
RESP score	0.5 (−2.5, 4.75)	−2 (−3.75, −1)	4.5 (3.5, 5)	0.02
HOSP. LOS PRE-ECMO (days)	6 (2.5, 13)	10 (6.5, 13.5)	2.5 (1.75, 3.5)	0.63
ICU LOS PRE-ECMO (days)	3 (2, 6.5)	6.5 (3.75, 7)	2.5 (1.75, 3.5)	0.04
MV PRE-ECMO (days)	3 (1.5, 6.5)	6.5 (3.75, 7)	2 (1, 3.5)	0.03
PaO_2_/FiO_2_ PRE-ECMO (mmHg)	61 (57.5, 70)	60 (56.25, 60.75)	67 (60, 70)	0.26
Creatinin (mg/dL)	1.36 (0.7, 3.05)	2.08 (0.94, 3.75)	1.3 (0.9, 2.02)	0.34
Hematocrit (%)	30.5 (30, 38.8)	30.1 (28.5, 30.42)	36.6 (32.4, 40.67)	0.38
Bilirubin (mg/dL)	0.92 (0.62, 1.26)	0.89 (0.79, 1.15)	1.15 (0.86, 1.30)	0.94
Acute Kidney Injury, N (%)	6 (54%)	4 (66%)	YES 2 (40%)	-
CRRT, N (%)	6 (54%)	3 (50%)	3 (60%)	-
Septic Shock, N (%)	10 (90%)	5 (83%)	5 (100%)	-
ECMO Duration (days)	22 (14, 39.5)	29.5 (11.75, 39.75)	25 (16.5, 39.5)	0.74
ECMO Survival, N (%)	10 (91%)	5 (83%)	5 (100%)	-
LOS ICU POST ECMO (days)	30.5 (19, 46.75)	38.5 (22, 46.75)	19.5 (8.75, 51.25)	0.81
Total Hospital LOS	60 (31.5, 104)	75.5 (38.25, 108.25)	35.5 (31.5, 59)	0.54

Continuous variables are presented as median value (25th to 75th percentile range) and nominal variables are presented as absolute quantity (percentage). BMI: body mass index; SAPS II Score: Simplified Acute Physiology 2 score; SOFA score; Sequential Organ Failure Assessment score; RESP score: Respiratory Extracorporeal Membrane Oxygenation Survival Prediction score; LOS HOSP: length of stay in hospital; ICU: intensive care unit; MV: mechanical ventilation; PaO_2_:FiO_2_: ratio of fraction of partial pressure of O_2_ to inspired O_2_; AKI: acute kidney injury; CRRT: continuous renal replacement therapy; ECMO: extracorporeal membrane oxygenation.

**Table 2 jpm-11-00015-t002:** Top ranked miRNAs and genes found in the network analysis.

Tissue Remodeling		Regulation of Immune System		Regulation of Coagulation	
TOP miRNAs	Genes	TOP miRNAs	Genes	TOP miRNAs	Genes
hsa-mir-20a-5p	223	hsa-mir-20a-5p	220	hsa-mir-20a-5p	9
hsa-mir-17-5p	218	hsa-mir-17-5p	194	hsa-mir-93-5p	9
hsa-mir-93-5p	177	hsa-mir-93-5p	169	hsa-let-7a-5p	8
hsa-mir-19b-3p	164	hsa-mir-106b-5p	159	hsa-mir-17-5p	7
hsa-mir-106b-5p	161	hsa-mir-19b-3p	131	hsa-mir-106b-5p	7
hsa-mir-126-5p	104	hsa-mir-126-5p	86	hsa-mir-19b-3p	7
hsa-mir-18a-5p	83	hsa-mir-18a-5p	74	hsa-mir-18a-5p	7
hsa-mir-92a-5p	49	hsa-mir-19a-5p	49	hsa-mir-126-5p	7
hsa-mir-19a-5p	45	hsa-mir-92a-5p	36	hsa-mir-99a-5p	7
hsa-mir-99a-5p	40	hsa-mir-99a-5p	33	hsa-mir-769-5p	7
hsa-mir-769-5p	33	hsa-mir-769-5p	29	hsa-mir-92a-5p	7
hsa-mir-212-5p	22	hsa-mir-212-5p	26	hsa-mir-212-5p	7

**Table 3 jpm-11-00015-t003:** Top ranked genes and miRNAs found in the network analysis.

Tissue Remodeling		Regulation of Immune System		Regulation of Coagulation	
TOP Genes	miRNAs	TOP Genes	miRNAs	TOP Genes	miRNAs
CCND1	9	CCND1	9	CCND1	10
CDKN1A	9	CDKN1A	9	CDKN1A	9
BTG2	9	BTG2	9	APP	9
APP	8	TNRC6A	9	BACH1	8
BMPR2	8	APP	8	ATP2A2	8
MDM2	8	BMPR2	8	ATP2B1	7
PMAIP1	8	MDM2	8	ACTB	6
ARHGAP5	7	PMAIP1	8	ACVR1B	5
ATP2B1	7	TNRC6B	8	SERPINA1	4
E2F1	7	PELI1	8	CDK6	4
EIF4G2	7	AGO3	8	PARP1	3
ELK4	7	CLTC	8		
GABBR1	7	ARHGAP5	7		
MAPK1	7	ATP2B1	7		
VEGFA	7	E2F1	7		
TNFRSF10B	7	EIF4G2	7		
MAP3K2	7	GABBR1	7		
SSX2IP	7	MAPK1	7		
SESN3	7	TXNIP	7		
CALM1	7	FRS2	7		

**Table 4 jpm-11-00015-t004:** Area under the curve (AUC) (95% confidence interval) for individual severity score of ARDS and for outcome indicators.

Marker	AUC	95% CI for AUC	*p*-Value
SAPS II	0.85	0.55–1	<0.05
SOFA	0.83	0.56–1	<0.05
RESP score	0.86	0.60–1	<0.001
ECMO DURATION	0.51	0.13–0.90	0.93
LOS ICU	0.76	0.42–1	0.12
LOS HOSP	0.53	0.15–0.91	0.86

**Table 5 jpm-11-00015-t005:** Area under the curve (AUC) (95% confidence interval) for individual miRNAs.

Marker	AUC	95% CI for AUC	*p*-Value
miR-93-5p	0.86	0.63–1	<0.01
miR-99a-5p	0.93	0.78–1	<0.01
miR-92a-5p	0.90	0.71–1	<0.01
miR-212-5p	0.90	0.68–1	<0.01
miR-20a-5p	0.93	0.78–1	<0.01
miR-19b-3p	0.93	0.78–1	<0.01
miR-18a-5p	0.88	0.68–1	<0.01
miR-17-5p	0.96	0.87–1	<0.01
miR-126-5p	0.90	0.68–1	<0.01
miR-106b-5p	0.80	0.40–1	0.13
miR-769-5p	0.86	0.63–1	<0.01
let-7a-5p	0.86	0.59–1	<0.01
miR-19a-5p	0.93	0.78–1	<0.01

## Data Availability

The datasets used and analyzed are available from the corresponding author on reasonable request.

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
