# Peer review of "Identification of a Circulating miRNA Signature to Stratify Acute Respiratory Distress Syndrome Patients"

_jpm, 2020, doi:10.3390/jpm11010015_

Round 1

Reviewer 1 Report

Micelli and colleagues presented an easy to understand preliminary study for differentially expressed microRNAs as a signature to stratify the severity of acute respiratory distress syndrome (ARDS). However, the authors have made some bold claims that these microRNAs can be used as biomarkers for ARDS. The science supporting these claims is not sufficient to warrant this term. These microRNAs need a lot more validation before identifying them as biomarkers. As this is a disease with no known treatment options, and little known about the causes and pathophysiology, I believe that with major revisions this paper will improve the scientific community’s understanding of ARDS and advance research in the field.

Major concerns:

  1. The technology used, TaqMan low density arrays are a good platform but has limitations. Had this study been done using RNA-seq, a lot more data perhaps have been obtained. We understand that performing an RNA-seq experiment is out of the scope of this paper, however, a more in-depth analysis of the existing data is required.
  2. The study is exceptionally small in cohort size and the authors rightly addressed this issue. The key requirements to claim the discovery of a biomarker are 1) the study needs to be done in a larger cohort, 2) the miRs need to be validated in independent cohorts from different locations, and 3) multiple techniques should be used at different locations with different cohorts analysed by different personnel. The use of the term biomarker is misleading, needs to be changed to signature as in the title.
  3. The study involves two clusters; however, a healthy control group is missing. Without a healthy control group, and showing a real difference in the miR profiles, how are we going to believe that these miRs are true potential biomarkers and not some artefacts?
  4. Overall 82% of the cohort were males. The authors need to include a cohort that is not gender-biased. Once again, we acknowledge that getting more patient samples is perhaps out of the scope of this paper, however, it can be discussed to reflect the gender difference in ARDS and the expected outcomes?
  5. The normalisation of the miR data is not what is generally done. A U6 normalisation followed by normalisation against one of the samples should suffice. Can the authors please renormalise against the housekeeping gene and then to cluster A or B for more mainstream normalisation?
  6. The introduction needs more explanation of the SAPS, SOFA, and RESP scores. The authors have explained them in one of the tables, but before getting to the table, the reader needs to know what these scores mean, why are Cluster A values mostly higher than Cluster B, except for RESP score, etc.
  7. There is a lot of variability in SOFA score for example (and in all the data), in the overall and cluster A, but not for cluster B. Can the authors comment on why there is so much variability?
  8. Line 296: MicroRNAs are associated with poor prognosis, given the variation within cluster A, and with cluster B, can the authors comment on any other factors or disorders that may explain elevated microRNA expression.
  9. The authors need to work a lot on the discussion. The miRs ultimately regulate proteins that control the pathways. Without any validation of protein data, how are supposed to know that these miRs are actually regulating the pathways mentioned by the authors? This is merely a prediction study. Without the protein data, big claims like “This focus is novel, and may determine a new class of biomarkers that could better define the prognosis, as well as be considered in evaluating treatments put in place”
  10. No discussion has been offered for why there is upregulation of microRNA in Fig 4A which is not observed in Fig 5 (ns). If within such a small study the authors fail to validate 13 miRs, what would eventually happen when other confounding variables kick in from different cohorts?

Minor concerns:

  1. The paper lacks consistency across terms and abbreviations. For example, in the methods section qRT-PCR is described, however, in results and figure legends RT-PCR is used.
  2. These sentences need to be rewritten.
    1. Line 55: ‘… as well as consume of coagulation factors and cells further stimulating the inflammatory and tissue repair pathways.
    2. Line 70: Why are these biomarkers peculiar? The whole rest of this paragraph needs clarification.
    3. Line 87: change since then to subsequently.
    4. Line 97: Spelling
    5. Line 98: change profiled by to profiled using.
    6. Line 106/107: U6 as a housekeeping gene.
    7. Line 111: common scale? This is normalisation. Needs to be described as such.
    8. Line 140: All confidence intervals are reported at 95%.
    9. Line 145-148: Exact copy of the methods. Is this required? Can you not just reference the methods.
    10. Line 169: remove the hyphen
    11. Fig2D: Cannot decipher the x-axis terms. Needs to be bigger.
    12. Line 180: Need to reference the individual methods section not just say ‘as in material and methods section’. (Section 2.2)
    13. Line 220: Why is Gene Ontology written in full but KEGG is abbreviated? Also, three lines down Gene Ontology is written in full again. Need to maintain consistency.
    14. Line 226: publicly available online applications
    15. Line 226: Consistent not consonant.
    16. Line 247: Expression is quite variable within cluster A, how can the microRNAs be discriminatory?
    17. Line 259: treatment the maneuvers?
    18. Line 290: prognostication?

Author Response

Reviewer 1

Miceli and colleagues presented an easy to understand preliminary study for differentially expressed microRNAs as a signature to stratify the severity of acute respiratory distress syndrome (ARDS). However, the authors have made some bold claims that these microRNAs can be used as biomarkers for ARDS. The science supporting these claims is not sufficient to warrant this term. These microRNAs need a lot more validation before identifying them as biomarkers. As this is a disease with no known treatment options, and little known about the causes and pathophysiology, I believe that with major revisions this paper will improve the scientific community’s understanding of ARDS and advance research in the field.

Response: Thank you very much for carefully reading the manuscript and your comments. We also appreciate the constructive critiques and suggestions you raised.

 Major concerns:

  1. The technology used, TaqMan low density arrays are a good platform but has limitations. Had this study been done using RNA-seq, a lot more data perhaps have been obtained. We understand that performing an RNA-seq experiment is out of the scope of this paper, however, a more in-depth analysis of the existing data is required.

Response: We appreciate your kind and important suggestion. We admit that RNa-seq investigation would lead to a more in-depth analysis of miRNAs profiles as well as discovery of miRNAs. But, as the reviewer said, in our work we aimed to conduct a preliminary screening of more highly characterized miRNAs. TaqMan low density arrays offers a simple and intuitive workflow to profile 754 well characterized miRNAs, and different groups showed this approach as a gold-standard for miRNA quantification (PMID: 30375585; 28837925; 29145074) widely used to identify miRNAs as biomarkers (PMID: 24339880; 31530881; 28641313; 27826919; 28275081). However, accordingly with the reviewer, we believe that our preliminary data revealed that different patterns of miRNA expression appear to be related to ARDS severity, and this leads to planning a more challenging analysis to analyze the miRNome in ARDS patients. 

  1. The study is exceptionally small in cohort size and the authors rightly addressed this issue. The key requirements to claim the discovery of a biomarker are 1) the study needs to be done in a larger cohort, 2) the miRs need to be validated in independent cohorts from different locations, and 3) multiple techniques should be used at different locations with different cohorts analysed by different personnel. The use of the term biomarker is misleading, needs to be changed to signature as in the title.

Response: As you suggested, we have deleted “biomarkers” and changed the text in the revised manuscript to avoid a misunderstanding for the reader (pag. 1, line 33; pag. 2, line 85; pag. 17, lines 312, 347, 353). For sure the study is small and, accordingly, we called it pilot, but we believe it adds at least the concrete idea that miRNAs may be worth being explored further. Currently, this method of investigation is quite expensive and complex, consequently our data can also suggest to other centers to follow this path.

  1. The study involves two clusters; however, a healthy control group is missing. Without a healthy control group, and showing a real difference in the miR profiles, how are we going to believe that these miRs are true potential biomarkers and not some artefacts?

Response:We thank the reviewer for the useful comments and will certainly think about the inclusion of healthy controls when planning the next experiments. In this phase of our project, considering the marked clinical and biologic heterogeneity within ARDS patients, we designed our study to an understanding of ARDS subphenotypes. Therefore, we tested whether the miRNA expressions were able to distinguish different ARDS subphenotypes in a heterogeneous group of ARDS patients. Actually, miRNAs have already been explored in critical care, and it would be really preliminary (but not impossible) to compare miRNAs in critically ill patients and healthy subjects. With future expansion of the project the different clusters will be better defined by inflammatory markers.

  1. Overall 82% of the cohort were males. The authors need to include a cohort that is not gender-biased. Once again, we acknowledge that getting more patient samples is perhaps out of the scope of this paper, however, it can be discussed to reflect the gender difference in ARDS and the expected outcomes?

Response: Dear Reviewer, thank you for pointing this out. We are preminently a clinical center and we work in the field. The real scene here is that ARDS patients are more predominately male in Sicily than female. Just as an example, in the first 30 patients admitted for Covid 19 at ISMETT just one was female. Is that because of obesity or smoking or whatever we can’t say. But we believe that in a consecutive cohort this is not really a gender bias. But your observation was correct, and we have added it as a future topic of investigation (pag 17, lines 359-360)

  1. The normalisation of the miR data is not what is generally done. A U6 normalisation followed by normalisation against one of the samples should suffice. Can the authors please renormalise against the housekeeping gene and then to cluster A or B for more mainstream normalisation?

Response: As you suggested, we normalize against the housekeeping gene for a more descriptive clustering and change Fig. 2a (pag. 7) and 4b (pag. 11) in the revised manuscript.

  1. The introduction needs more explanation of the SAPS, SOFA, and RESP scores. The authors have explained them in one of the tables, but before getting to the table, the reader needs to know what these scores mean, why are Cluster A values mostly higher than Cluster B, except for RESP score, etc.

Response: Dear reviewer thank you for suggesting that. Actually you are right on that and give us the opportunity to better explain in the Introduction the scores (pag. 2, lines 51-57). Moreover, we have also explained why the Resp score has different values in the Methods and discussed it more (pag 3, lines 99-102).

  1. There is a lot of variability in SOFA score for example (and in all the data), in the overall and cluster A, but not for cluster B. Can the authors comment on why there is so much variability?

Response: We really believe that this is a random effect. Actually, probably in a small cohort the SOFA, when it has higher values may be more variable. But consider that for critically ill patients a median of 5 is not that high. Considering that in ECMO patients we tend to exclude patients with compromised multiple organ failure.

  1. Line 296: MicroRNAs are associated with poor prognosis, given the variation within cluster A, and with cluster B, can the authors comment on any other factors or disorders that may explain elevated microRNA expression.

Response: We have changed this sentence, and now it is probably more adequate to the previous topic (pag 17, lines 345-346).

  1. The authors need to work a lot on the discussion. The miRs ultimately regulate proteins that control the pathways. Without any validation of protein data, how are supposed to know that these miRs are actually regulating the pathways mentioned by the authors? This is merely a prediction study. Without the protein data, big claims like “This focus is novel, and may determine a new class of biomarkers that could better define the prognosis, as well as be considered in evaluating treatments put in place”

Response: We thank the reviewer for the useful comments. Our in silico analysis of the 13 deregulated miRNAs was obtained using miRNET database. MiRNET is an easy-to-use Web-based tool that offers statistical, visual, and network-based approaches to help researchers understand miRNA functions and regulatory mechanisms. When compared with other tools as DIANA-miRPath, starBase-miRFunction, miRTar and miRSystem, miRNet offers a unique set of features with regard to statistical data analysis, network visualization support, and functional interactions. The key features of miRNet include a comprehensive knowledge base integrating high-quality miRNA-target interaction data (including validated data) from 11 databases (miRTarBase, TarBase, miRecords, SM2miR, Pharmaco-miR, miR2Disease, PhenomiR, StarBase, EpimiR, miRDB and miRanda) (PMID: 27105848). Our data summarized in Fig. 3, Tables 2 and 3, are the result of this kind of analysis. Moreover, in order to answer the reviewer’s request we have worked on the Discussion, adding some papers that show validated correlations between miRNAs and targets (pag. 17, lines 330-334).

  1. No discussion has been offered for why there is upregulation of microRNA in Fig 4A which is not observed in Fig 5 (ns). If within such a small study the authors fail to validate 13 miRs, what would eventually happen when other confounding variables kick in from different cohorts?

Response: We appreciate this comment.We believe that this is due to the low number of patients, which leads to high variability of the expression of these miRNAs in the two groups. We believe that if these miRNAs were candidates to be biomarkers for ARDS, more patients could reduce their variability. Therefore, their different results would be statistically significant if we analyzed the average expression between the two groups with the Student's T test, while it is not significant if the expression data are compared with the Mann - Whitney U test. However, we discuss this in the revised manuscript (pag. 17, lines 335-337).

Minor concerns:

  1. The paper lacks consistency across terms and abbreviations. For example, in the methods section qRT-PCR is described, however, in results and figure legends RT-PCR is used.

Response: Thank you very much for carefully reading the manuscript. We have changed all “RT-PCR” in “qRT-PCR” in the revised manuscript.

  1. These sentences need to be rewritten.

Response: As you suggested, we have rewriten the following sentences in the revised manuscript.

    1. Line 55: ‘… as well as consume of coagulation factors and cells further stimulating the inflammatory and tissue repair pathways.

DONE: pag. 2 lines 62-63

    1. Line 70: Why are these biomarkers peculiar? The whole rest of this paragraph needs clarification.

DONE: pag. 2 lines 78-81

    1. Line 87: change since then to subsequently.

DONE: pag. 3 line 96

    1. Line 97: Spelling

DONE: pag. 3 line 110

    1. Line 98: change profiled by to profiled using.

DONE: pag. 3 line 111

    1. Line 106/107: U6 as a housekeeping gene.

DONE: pag. 3 line 119

    1. Line 111: common scale? This is normalisation. Needs to be described as such.

DONE: pag. 3 lines 121-124

    1. Line 140: All confidence intervals are reported at 95%.

DONE: pag. 4 line 155

    1. Line 145-148: Exact copy of the methods. Is this required? Can you not just reference the methods.

DONE: pag. 4 lines 161-164

    1. Line 169: remove the hyphen

DONE: pag. 5 line 186

    1. Fig2D: Cannot decipher the x-axis terms. Needs to be bigger.

DONE

    1. Line 180: Need to reference the individual methods section not just say ‘as in material and methods section’. (Section 2.2)

DONE: pag. 7 lines 199

    1. Line 220: Why is Gene Ontology written in full but KEGG is abbreviated? Also, three lines down Gene Ontology is written in full again. Need to maintain consistency.

DONE

    1. Line 226: publicly available online applications

DONE: pag. 12 line 254 

    1. Line 226: Consistent not consonant.

DONE: pag. 12 line 255 

    1. Line 247: Expression is quite variable within cluster A, how can the microRNAs be discriminatory?

DONE: pag. 16 line 283.

    1. Line 259: treatment the maneuvers?

DONE: pag. 16 line 297.

    1. Line 290: prognostication?

DONE: pag. 17 line 339.

Reviewer 2 Report

In this manuscript the Authors verified if miRNA can identify two different groups of ARDS patient. The topic is interesting and well presented. The title is informative, methods well reported and conclusions are consistent with results.

Comments:

- Even if there is not a structured abstract, the number of enrolled patients should be moved after the methods’ description. Clarify what is V-V

- The meaning of sentence in L57-58 is not clear.

- A brief recap about SAPS II e SOFA would be welcomed.

- It is not clear if the Authors decided a priori to identify two clusters or if the miRNA analysis showed only two subgroups.

- Move L145-8 in methods. 

- In order to test if the number of patients is sufficient I suggest to use notched box-and-whiskers plots

Author Response

Reviewer 2

In this manuscript the Authors verified if miRNA can identify two different groups of ARDS patient. The topic is interesting and well presented. The title is informative, methods well reported and conclusions are consistent with results.

Response: We very much appreciate the time and effort of the reviewer. Thank you very much for carefully reading the manuscript and your comments.

Comments:

- Even if there is not a structured abstract, the number of enrolled patients should be moved after the methods’ description. Clarify what is V-V

Response: As you suggested, we have moved the number of enrolled patients after the Methods and clarified what is meant by V-V (pag. 1 lines 20, 21, 26)

- The meaning of sentence in L57-58 is not clear.

Response: we have synthesized this sentence and we believe it is now more understandable (pag 2, lines 62-63)

- A brief recap about SAPS II e SOFA would be welcomed.

Response: We have done this in the Introduction, and also better described the values in the Methods (pag 2 lines 51-57 and pag 3, lines 99-102)

- It is not clear if the Authors decided a priori to identify two clusters or if the miRNA analysis showed only two subgroups.

Response: We apologize to the reviewer if we have not explained this well. We identified two clusters by hierarchical clustering analysis on filtered miRNAs (quality control screening) as described on page 5, line 173-176.

- Move L145-8 in methods. 

Response: Thank you very much for carefully reading the manuscript. We have moved the sentence in Materials and Methods section (pag 2 lines 91-94)

- In order to test if the number of patients is sufficient I suggest to use notched box-and-whiskers plots

Response: We thank the reviewer for the useful comment. We show the data with box and whiskers plot and have changed Figure 5
